# Hydrodenitrogenation of pyridines and quinolines at a multinuclear titanium hydride framework

Shaowei Hu[1], Gen Luo [1,2], Takanori Shima [1], Yi Luo[2] & Zhaomin Hou [1,2]

Investigation of the hydrodenitrogenation (HDN) of aromatic N-heterocycles such as pyridines and quinolines at the molecular level is of fundamental interest and practical importance, as this transformation is essential in the industrial petroleum refining on solid catalysts. Here, we report the HDN of pyridines and quinolines by a molecular trinuclear titanium polyhydride complex. Experimental and computational studies reveal that the denitrogenation of a pyridine or quinoline ring is easier than the ring-opening reaction at the trinuclear titanium hydride framework, which is in sharp contrast with what has been reported previously. Hydrolysis of the pyridine-derived nitrogen-free hydrocarbon skeleton at the titanium framework with $H_2O$ leads to recyclization to afford cyclopentadiene with the generation of ammonia, while treatment with HCl gives the corresponding linear hydrocarbon products and ammonium chloride. This work has provides insights into the mechanistic aspects of the hydrodenitrogenation of an aromatic N-heterocycle at the molecular level.

[1] RIKEN Center for Sustainable Resource Science and Organometallic Chemistry Laboratory, RIKEN, 2-1 Hirosawa, Wako, Saitama 351-0198, Japan. [2] State Key Laboratory of Fine Chemicals, School of Chemical Engineering, Dalian University of Technology, Dalian 116024, China. Shaowei Hu and Gen Luo contributed equally to this work. Correspondence and requests for materials should be addressed to Y.L. (email: luoyi@dlut.edu.cn) or to Z.H. (email: houz@riken.jp)

The hydrodenitrogenation (HDN) of aromatic N-heterocycles such as pyridines and quinolines is an important process in the industrial petroleum refining to remove nitrogenous impurities from crude oil[1–4]. This transformation is essential not only to suppress $NO_x$ emissions upon combustion of the fuel but also to improve the performance of the hydrocracking and other downstream processes. HDN is now gaining more importance with the move toward alternative natural resources such as biomass and shale, which contain higher nitrogen contents. The C–N bonds of aromatic N-heterocycles are quite stable and are therefore difficult to break under ordinary conditions[5,6]. The industrial HDN is carried out at high temperatures (300–500 °C) and high pressures (up to 200 atm) on the surfaces of solid catalysts such as NiMo/$Al_2O_3$ and CoMo/$Al_2O_3$[1–4]. To aid the design of new catalysts and achieve HDN under milder conditions, a better understanding of the reaction mechanism is of significant importance. It was generally believed that the elementary steps in the industrial HDN process might include sequentially N-heterocycle coordination to surface metal sites, aromatic ring hydrogenation, ring opening through C–N bond cleavage to give a linear amine, and denitrogenation to yield nitrogen-free linear hydrocarbons with the release of ammonia (see Fig. 1)[1,2]. However, because of the complexity of the solid catalysts, which may contain many different types of active sites, further clarification of the operative reaction mechanism at the molecular level has remained a formidable challenge to date.

Investigation of the reactions of aromatic N-heterocycles with molecular organometallic complexes has been expected to bring about a better understanding of the industrial HDN reaction mechanism and thereby lead to the development of more efficient transformations under milder conditions[7–21]. However, studies in this area have met with only limited success to date. A wide range of transition metal complexes have been found to bind pyridine and quinoline in various coordination modes without causing C–N bond cleavage[7–12]. The combination of some organometallic complexes with strongly reducing agents or bases was found to induce the ring opening of pyridine units through cleavage of one of the two C–N bonds, but extrusion of the nitrogen atom by cleavage of the remaining C–N bond in a ring-opening product was not observed[13–19]. When treated with $Me_3SiCl$, the denitrogenation of a pyridine ring-opening product generated by metathesis with a titanium alkylidyne species was achieved[20–22]. The denitrogenation of a fused aromatic N-heterocycle such as quinoline by a molecular organometallic complex has not been reported previously[23].

In view of the fact that the industrial HDN process might involve transition metal hydrides as the true active species, investigation of the reactions of molecular transition metal hydrides with aromatic N-heterocycles is especially of interest and importance, as this approach may provide a useful entry into homogeneous HDN systems. Toward this end, a number of transition metal hydride complexes have been examined with aromatic N-heterocycles such as pyridines and quinolines, and various reaction patterns such as coordination, C–H activation, hydrogenation, hydroboration, and hydrosilylation have been observed[24–37]. However, the denitrogenation of an aromatic

N-heterocycle by a molecular metal hydride has remained unknown to date.

We have previously shown that multimetallic titanium polyhydride complexes such as $[(C_5Me_4SiMe_3)Ti]_3(\mu_3\text{-}H)(\mu_2\text{-}H)_6$ (1) exhibit an unusually high activity for the activation of some very stable chemical bonds, such as cleaving dinitrogen ($N_2$) and a benzene ring at room temperature[38–42]. These results promoted us to examine whether titanium hydride clusters could induce the ring opening and denitrogenation of an aromatic N-heterocycle. Here, we report our studies on the reaction of the trinuclear titanium heptahydride complex $[(C_5Me_4SiMe_3)Ti]_3(\mu_3\text{-}H)(\mu_2\text{-}H)_6$ (1) with pyridines and quinolines. We have found that the nitrogen atom in a pyridine or quinoline ring can be extruded under mild conditions at the trinuclear titanium framework through reduction of a HC=N unit followed by cleavage of the two C–N bonds. The mechanistic aspects have been elucidated by computational studies and the isolation and identification of some key reaction intermediates. Both linear and cyclic nitrogen-free hydrocarbon products have been obtained from pyridine by choosing the appropriate protonolysis conditions.

## Results

**Denitrogenation of pyridines.** When pyridine was mixed with the trinuclear titanium heptahydride complex $[(C_5Me_4SiMe_3)Ti]_3(\mu_3\text{-}H)(\mu_2\text{-}H)_6$ (1)[38] in hexane at room temperature (23–25 °C), an immediate color change from dark green to dark purple was observed. Complex $[(C_5Me_4SiMe_3)Ti]_3(\mu\text{-}\eta^1{:}\eta^2{:}\eta^2\text{-}C_5H_4N)(\mu_2\text{-}H)_3(\mu_3\text{-}H)$ (2) was formed and isolated as dark-purple crystals in 93% yield (Fig. 2). The release of $H_2$ was observed at δ 4.54 by the [1]H NMR (nuclear magnetic resonance) monitoring of the reaction in deuterated tetrahydrofuran (THF-$d_8$). No intermediate species was observed even when the reaction was carried out at a low temperature (−30 °C), suggesting that the transformation of 1 and pyridine to 2 is quite facile.

An X-ray diffraction study revealed that 2 possesses a $Ti_3(\mu_2\text{-}H)_3(\mu_3\text{-}H)$ core structure. A pyridyl group ($C_5H_4N$) is bonded in a $\mu\text{-}\eta^1{:}\eta^2{:}\eta^2$-manner to the three Ti atoms via a C–N unit (C5–N1). The carbon atom (C5) directly bonded to the titanium framework does not have a hydrogen atom. The bond distance of the Ti-interacted C5–N1 bond (1.427(4) Å) is significantly longer than that of the metal-free C9–N1 bond (1.393(4) Å). The Ti1···Ti2 distance (2.6516(8) Å) is slightly shorter than those of Ti1···Ti3 (2.6817(8) Å) and Ti2···Ti3 (2.6854(8) Å). Formally, the $C_5H_4N$ moiety in 2 bears three negative charges, one being on the N1 atom and two on the C5 atom. Similar to 1, one of the three Ti atoms in 2 could formally be viewed as the 4+ oxidation state Ti(IV) and two atoms could be viewed as the 3+ oxidation state Ti(III), in consideration of the total negative charges of all the ligands on the titanium atoms. In this transformation, three of the seven hydride ligands in 1 were consumed, one being used for the ortho-C–H bond activation (deprotonation) of pyridine, and two being released as $H_2$ by donating two electrons. Therefore, this could account for the formation of the trianionic $[C_5H_4N]^{3-}$ species in 2.

Analogously, the reaction of pyridine-$d_5$ with 1 afforded $[(C_5Me_4SiMe_3)Ti]_3(\mu\text{-}\eta^1{:}\eta^2{:}\eta^2\text{-}C_5D_4N)(\mu_2\text{-}H)_3(\mu_3\text{-}H)$ (2-$d_4$) with the release of $H_2$ and HD. The [1]H or [2]H NMR spectrum of 2-$d_4$ did not show H/D scrambling. The reaction of 1 with [15]N-enriched pyridine yielded $[(C_5Me_4SiMe_3)Ti]_3(\mu\text{-}\eta^1{:}\eta^2{:}\eta^2\text{-}C_5H_4{}^{15}N)(\mu_2\text{-}H)_3(\mu_3\text{-}H)$ (2-[15]N), which showed a singlet at δ −122.7 in the [15]N NMR spectrum.

When 2 was heated in hexane at 60 °C for 12 h, a nitride/pentadienyl complex $[(C_5Me_4SiMe_3)Ti]_3[\mu\text{-}\eta^2{:}\eta^2{:}\eta^1{:}\eta^1\text{-}(CH)_5](\mu_3\text{-}N)(\mu_2\text{-}H)$ (3) was obtained in 77% yield as dark-green crystals (Fig. 2). Heating 2-[15]N at 60 °C afforded the [15]N-enriched analog

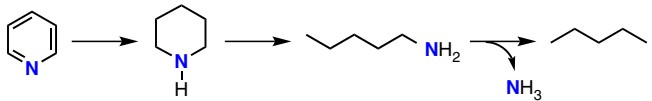

**Fig. 1** Some key steps proposed for the HDN of pyridine on solid catalysts. The reaction may proceed via pyridine coordination, ring hydrogenation, ring opening, and denitrogenation

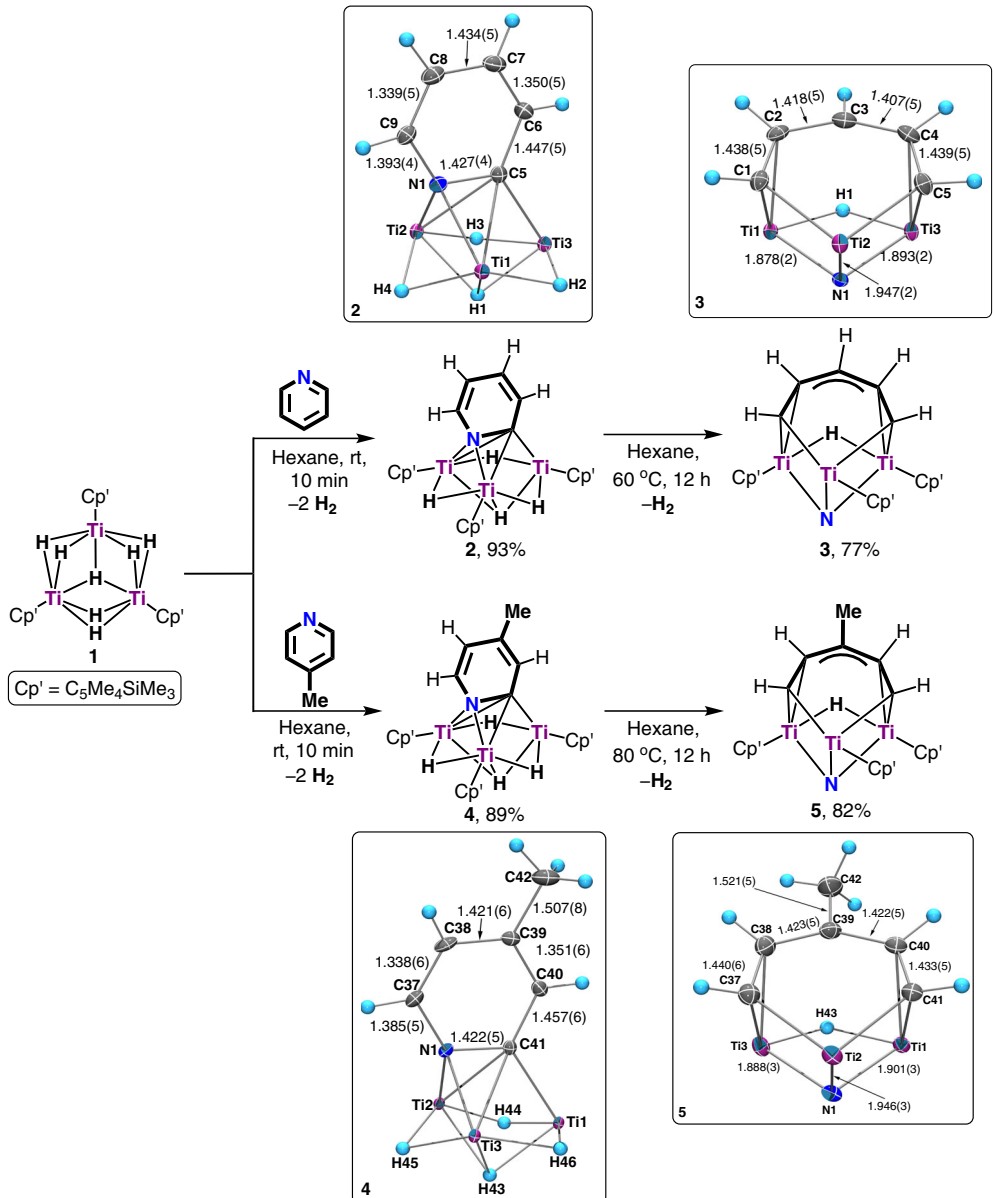

**Fig. 2** Hydrodenitrogenation of pyridines by the titanium hydride complex **1**. Reactions of **1** with pyridine and 4-methylpyridine at room temperature gave **2** and **3**, which upon heating at 60–80 °C extruded the nitrogen atom from the pyridine unit to give **4** and **5**, respectively (Cp′ = C₅Me₄SiMe₃). The X-ray structures of **2**, **3**, **4**, and **5** at 30% probability ellipsoids are shown in the square frames, with the C₅Me₄SiMe₃ ligands being omitted for clarity. The bond distances of **2**, **3**, **4**, and **5** are given in angstrom (Å)

$[(C_5Me_4SiMe_3)Ti]_3[\mu\text{-}\eta^2\text{:}\eta^2\text{:}\eta^1\text{:}\eta^1\text{-}(CH)_5](\mu_3\text{-}^{15}N)(\mu_2\text{-}H)$ (**3-$^{15}$N**), which showed a singlet at δ 469.3 in the $^{15}$N NMR spectrum, which was in agreement with the formation of a μ-N nitrido titanium species[38,43,44]. Complex **3** could also be obtained directly by the reaction of **1** with pyridine at 60 °C for 12 h. An X-ray diffraction study revealed that **3** is formally composed of a $[(C_5Me_4SiMe_3)_3Ti_3(\mu_3\text{-}N)(\mu_2\text{-}H)]^{5+}$ unit and a linear $[(CH)_5]^{5-}$ unit. One (Ti2) of the three Ti atoms is bonded to the two terminal carbon atoms (C1 and C5) of the $(CH)_5$ group to form a six-membered metallacyle, and the other two Ti atoms (T1 and Ti3) are bonded to two different terminal C−C units of the $(CH)_5$ group in a $\eta^2$ manner. The Ti−C bond distances within the metallacycle (Ti2−C1: 2.063(3) Å, Ti2−C5: 2.051(3) Å) are much shorter than those of the outside $\eta^2$-Ti−C bonds (Ti1−C1: 2.201(3) Å, Ti1−C2: 2.242(3) Å, Ti3−C5: 2.210(3) Å, and Ti3−C4: 2.243(3) Å). The nitride atom (N1) is bonded to all three Ti atoms, with the Ti2−N1 bond distance (1.947(2) Å) being

significantly longer than those of the Ti1−N1 (1.878(2) Å) and Ti3−N1 (1.893(2) Å) bonds, but is comparable with those of the Ti−(μ₃-N) bonds found in $[(C_5Me_4SiMe_3)Ti]_3(\mu_3\text{-}N)(\mu_2\text{-}NH)(\mu_2\text{-}H)_2$ (1.853(5)−2.071(5) Å)[38]. The Ti···Ti distances (Ti1···Ti2: 2.9268(7), Ti1···Ti3: 2.8173(7), and Ti2···Ti3: 2.9355(8) Å) in **3** are much longer than those in **2** (2.6516(8)−2.6854(8) Å).

In the present transformation of **2** to **3**, the nitrogen atom was extruded from the $[C_5H_4N]^{3-}$ moiety together with partial hydrogenation of the hydrocarbon skeleton to generate a linear $(CH)_5$ unit. Two C−N bonds were cleaved and one new C−H bond was formed in this reaction. Three of the four hydride ligands in **2** were consumed, one being transferred to the $C_5$ species and two being released as $H_2$ to donate two electrons. Besides, the two Ti(III) sites in **2** were formally oxidized to Ti(IV) in **3** to release two electrons. The total four electrons generated in this way together with the hydride transfer from titanium would account for the transformation (reduction and C−N bond

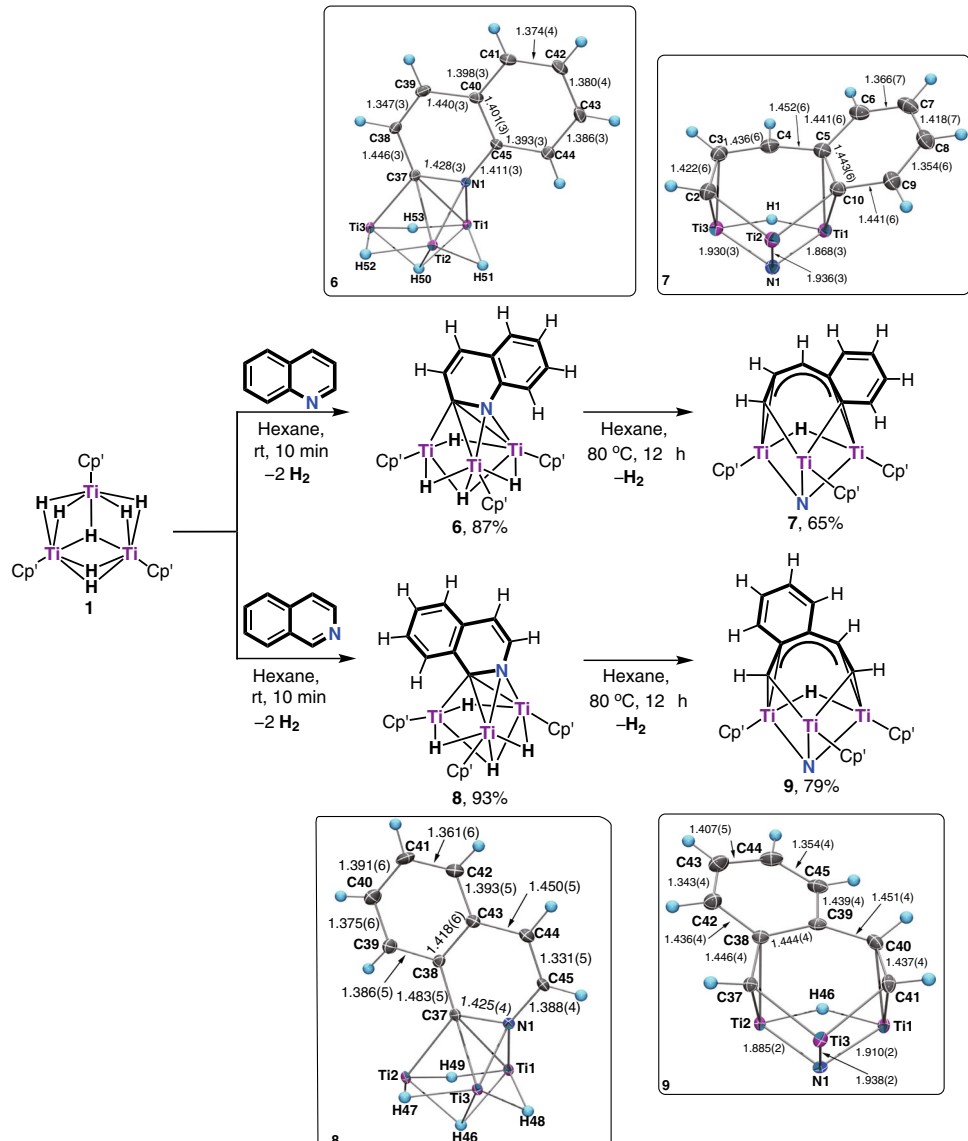

**Fig. 3** Hydrodenitrogenation of quinoline and isoquinoline by the titanium hydride complex **1**. Reactions of **1** with quinoline and isoquinoline at room temperature gave **6** and **8**, which upon heating at 80 °C extruded the nitrogen atom from the pyridine unit to give **7** and **9**, respectively. The X-ray structures of **6**, **7**, **8**, and **9** at 30% probability ellipsoids are shown in the square frames, with the $C_5Me_4SiMe_3$ ligands being omitted for clarity. The bond distances of **6**, **7**, **8**, and **9** are given in angstrom (Å)

cleavage) of the $[C_5H_4N]^{3-}$ species in **2** to the $[(CH)_5]^{5-}$ and $N^{3-}$ units in **3**.

Monitoring the conversion of **2** to **3** via $^1H$ NMR spectroscopy at room temperature did not show the formation of any intermediates. This may suggest that the denitrogenation step could be relatively easy after ring opening (cleavage of one C–N bond) of the $[C_5H_4N]$ unit (*vide infra*). These results stand in sharp contrast with what was previously observed in the reaction of pyridine with a titanium alkylidyne species, where denitrogenation was much more difficult than ring opening[22]. The kinetic studies of the conversion of **2** to **3** at a temperature range of 40–60 °C gave the activation parameters of $\Delta H^{\neq} = 25.3(5)$ kcal/mol, $\Delta S^{\neq} = -0.6(14)$ eu, and $\Delta G^{\neq}_{(298.15\ K)} = 25.5$ kcal/mol, suggesting that the rate-determining step should be a unimolecular process. The reaction rate was not significantly affected by using **2** vs. **2-$d_4$** ($k = 1.30 \times 10^{-4}\ s^{-1}$, $k_D = 1.09 \times 10^{-4}\ s^{-1}$) or under a $H_2$ atmosphere ($k_{(H2)} = 1.05 \times 10^{-4}\ s^{-1}$).

In an analogous manner, 4-methylpyridine was hydrodenitrogenated to $[(C_5Me_4SiMe_3)Ti]_3[\mu-\eta^2:\eta^2:\eta^1:\eta^1-(CH)_2C(Me)$

$(CH)_2](\mu_3-N)(\mu_2-H)$ (**5**) via $[(C_5Me_4SiMe_3)Ti]_3(\mu-\eta^1:\eta^2:\eta^2-C_9H_3MeN)(\mu_2-H)_3(\mu_3-H)$ (**4**) (Fig. 2), though a slightly higher temperature (80 °C) was needed to complete the reaction in 12 h. Both **4** and **5** were structurally characterized by X-ray diffraction and NMR analyses.

**Denitrogenation of quinolines.** Quinolines are typical fused aromatic *N*-heterocycles existing in crude oil, which are usually more difficult to activate than pyridines. The denitrogenation of a quinoline ring by a molecular metal complex has remained unknown to date[23]. We found that the titanium polyhydride complex **1** rapidly reacted with quinoline at room temperature, which gave the dehydrogenative reduction product $[(C_5Me_4SiMe_3)Ti]_3(\mu-\eta^1:\eta^2:\eta^2-C_9H_6N)(\mu_2-H)_3(\mu_3-H)$ (**6**) (an analog of **2** and **4**) in 87% yield within 10 min (Fig. 3). When **6** was heated in hexane at 80 °C for 12 h, the denitrogenation reaction took place to give $[(C_5Me_4SiMe_3)Ti]_3(\mu-\eta^2:\eta^2:\eta^1:\eta^1-CHCHCHC_6H_4)(\mu_3-N)(\mu_2-H)$ (**7**) by cleavage of the two C–N bonds. An X-ray diffraction study revealed that **7** is formally

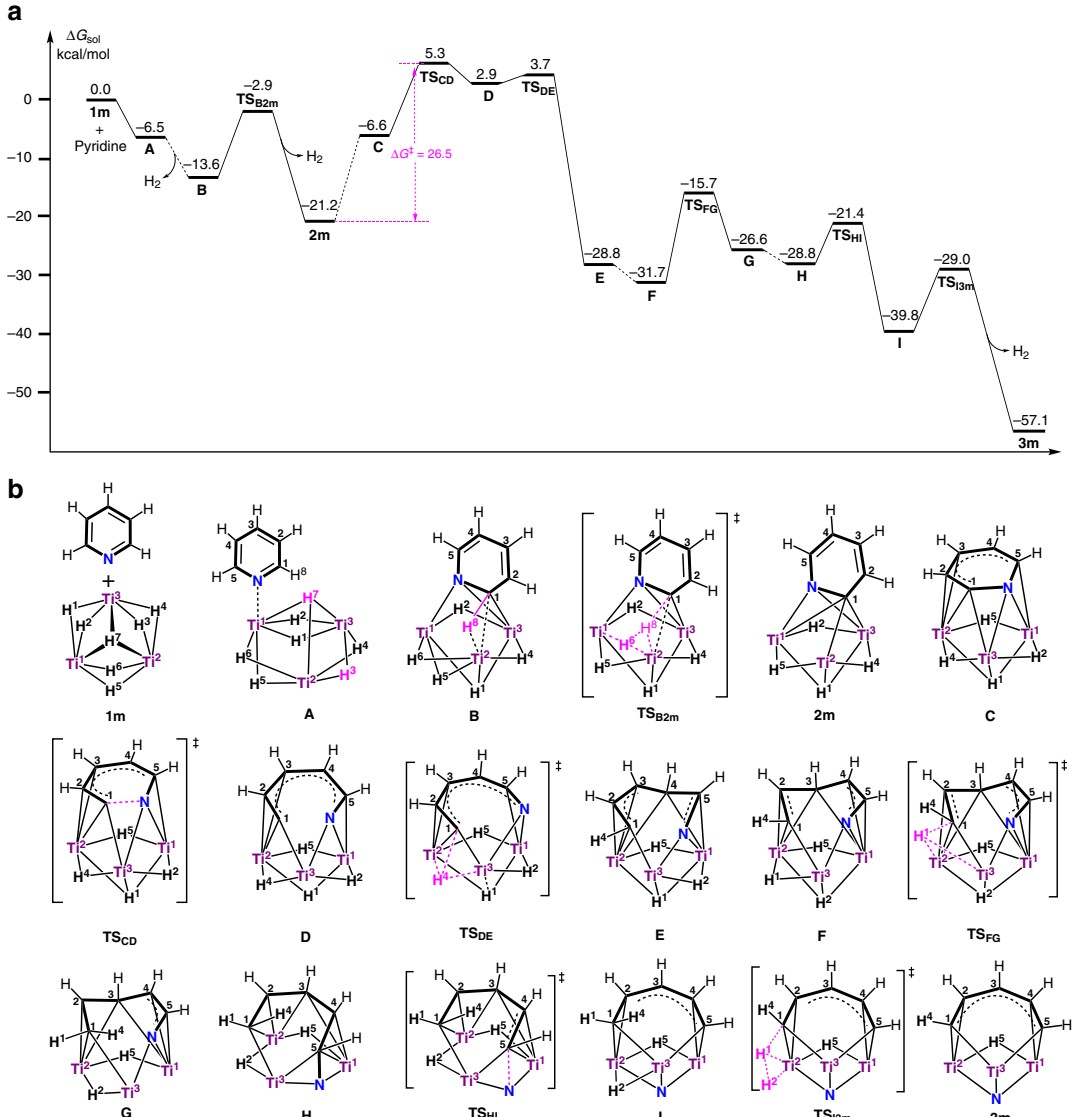

**Fig. 4** Computational analysis of the reaction of **1** with pyridine. **a** Simplified energy profile for the reaction of **1m** (a model of **1**) with pyridine. **b** Structures of the stationary points shown in the energy profile. The $C_5H_4SiH_3$ ligands have been omitted for clarity. The Gibbs free energies (kcal/mol) in solution are relative to the energy sum of **1m** and pyridine. The complex name with **TS** means transition state. The dashed line connected to two stationary points in the energy profile means that the detailed transformations involved in the corresponding steps have been omitted for clarity and are given in Supplementary Figs. 58 and 59

composed of a $[(C_5Me_4SiMe_3)_3Ti_3(\mu_3\text{-N})(\mu_2\text{-H})]^{5+}$ unit and a $[CHCHCHC_6H_4]^{5-}$ unit. One titanium atom (Ti2) is bonded to the terminal carbon atom (C2) of the $-(CH)_3-$ unit and an ortho carbon atom (C10) of the phenyl ring to form a six-membered metallacycle, while the other two titanium atoms (Ti1 and Ti3) are bonded, respectively, to the terminal C2–C3 moiety of the $-(CH)_3-$ unit and the C10–C5 unit of the phenyl ring in a $\eta^2$ manner (Fig. 3). Similar to **3** and **5**, the Ti–C bond distances within the metallacycle (Ti2–C10: 2.148(4) Å, Ti2–C2: 2.049(4) Å) are much shorter than those of the outside $\eta^2$-Ti–C bonds (Ti3–C2: 2.223(4) Å, Ti3–C3: 2.221(4) Å, Ti1–C5: 2.358(4) Å, and Ti1–C10: 2.289(4) Å). The nitride atom (N1) is bonded to three Ti atoms in a $\mu_3$-manner. The denitrogenation of iso-quinoline was also achieved analogously by reaction with **1**, which sequentially afforded the structurally characterizable dehydrogenative reduction product $[(C_5Me_4SiMe_3)Ti]_3(\mu\text{-}\eta^1{:}\eta^2{:}\eta^2\text{-iso-}C_9H_6N)(\mu_2\text{-H})_3(\mu_3\text{-H})$ (**8**) and the denitrogenation product $[(C_5Me_4SiMe_3)Ti]_3(\mu\text{-}\eta^2{:}\eta^2{:}\eta^1{:}\eta^1\text{-CHCHC}_6H_4CH)(\mu_3\text{-N})(\mu_2\text{-H})$

(**9**) in high yields (Fig. 3). It is also worth noting that the deprotonative reduction reaction took place regiospecifically at the most hindered site, probably due to electronic influence.

**Theoretical calculations**. The formation of **3**, **5**, **7**, and **9** from the reactions of the titanium hydride cluster **1** with pyridines and quinolines represents a unique example of denitrogenation of an aromatic *N*-heterocycle by a well-defined molecular metal hydride complex. To have a better understanding about the denitrogenation of pyridine by **1**, we performed the density-functional theory (DFT) calculations on a model complex of **1**, $[(C_5H_4SiH_3)Ti]_3(\mu_3\text{-H})(\mu_2\text{-H})_6$ (**1 m**), in which the $C_5Me_4SiMe_3$ ligand in **1** was replaced with $C_5H_4SiH_3$[40]. Some important mechanistic steps are shown in Fig. 4. The coordination of pyridine via the N atom to Ti1 in **1 m** gives $[(C_5H_4SiH_3)Ti]_3(\eta^1\text{-}NC_5H_5)(\mu_3\text{-H})(\mu_2\text{-H})_6$ (**A**), which then induces $H_2$ release (reductive elimination of H3 and H7) to afford $[(C_5H_4SiH_3)Ti]_3(\mu\text{-}\eta^1{:}\eta^2\text{-}C_5H_5N)(\mu_2\text{-H})_4(\mu_3\text{-H})$ (**B**). This process is

**Fig. 5** Hydrolysis reactions of **3** and **5** with HCl and $H_2O$. The reaction of **3** and **5** with HCl yielded the linear nitrogen-free hydrocarbons with the release of $NH_4Cl$, while the reaction with $H_2O$ afforded the cyclic hydrocarbon products with the release of $NH_3$

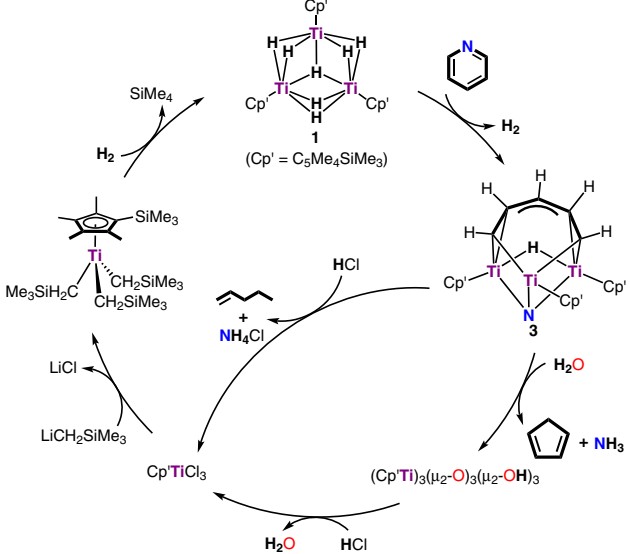

**Fig. 6** Titanium-mediated cycles for the hydrodenitrogenation of pyridine by $H_2$. The reaction of pyridine with **1** gave **3** with the release of $H_2$. Treatment of **3** with HCl afforded 1-pentene, $NH_4Cl$, and Cp'TiCl$_3$, while the reaction of **3** with $H_2O$ yielded cyclopentadiene, $NH_3$, and (Cp'Ti)$_3$($\mu_2$-O)$_3$($\mu_2$-OH)$_3$ which upon treatment with HCl gave Cp'TiCl$_3$. The reaction of Cp'TiCl$_3$ with LiCH$_2$SiMe$_3$ yielded Cp'Ti(CH$_2$SiMe$_3$)$_3$, which upon hydrogenolysis with $H_2$ regenerated **1**

accompanied by reduction of the C1 = N double bond in the pyridine moiety to the C1−N single bond by the low-valent titanium species (or electrons) generated by $H_2$ release. The deprotonative C1−H8 bond activation by the hydride ligand H6 in **B** via transition state $TS_{B2m}$ would yield [((C$_5$H$_4$SiH$_3$)Ti]$_3$($\mu$-$\eta^1$: $\eta^2$:$\eta^2$-C$_5$H$_4$N)($\mu_2$-H)$_3$($\mu_3$-H) (**2 m**) with the release of one molecule of $H_2$ (H6−H8). Complex **2 m** is equivalent to **2** obtained experimentally. The whole process of the transformation of **1 m** plus pyridine to **2 m** plus 2 $H_2$ overcomes an energy barrier of 13.8 kcal/mol (Supplementary Fig. 58) and is exergonic by 21.2 kcal/mol. Transformation of the [C$_5$H$_4$N] moiety in **2 m** from perpendicular to parallel to the Ti1–Ti2–Ti3 plane could take place to give a more reactive intermediate **C**, in which the oxidative addition of the C1−N bond in the [C$_5$H$_4$N] unit to the Ti3 atom via transition state $TS_{CD}$ (*i.e.*, C1−N bond cleavage) subsequently occurs to yield the ring-opening species **D**. The formal reductive elimination of the C1 atom and the hydride H4 in **D** via transition state $TS_{DE}$ (*i.e.*, C1−H4 bond formation) would give the thermodynamically stable complex **E**, which then easily isomerizes to **F** by rearrangement of the hydride ligands. The reductive elimination of C1 and H1 in **F** via transition state $TS_{FG}$ (i.e., C1−H1 bond formation) could afford an intermediate

**G**, which then changes to **H** by deformation of the C5−N bond accompanied by Ti3−C5 bond formation. The oxidative addition of the C5−N bond to Ti3 in **H** may take place via transition state $TS_{HI}$ to give **I**, which upon deprotonative C1−H1 bond activation by the adjacent hydride H2 (*i.e.*, H1−H2 formation) yields the more stable complex [(C$_5$H$_4$SiH$_3$)Ti]$_3$[$\mu$-$\eta^2$:$\eta^2$:$\eta^1$:$\eta^1$-(CH)$_5$]($\mu_3$-N) ($\mu_2$-H) (**3 m**) with the release of $H_2$. Complex **3 m** is equivalent to **3** obtained experimentally. The whole process of the transformation of **1 m** plus pyridine to **3 m** plus 3 $H_2$ is exergonic by 57.1 kcal/mol. The rate-determining step is the cleavage of the C1−N bond (ring opening) with an energy barrier of 26.5 kcal/mol (see also Supplementary Fig. 59), which is in good agreement with the experimental result ($\Delta G^{\neq}_{(298.15\ K)} = 25.5$ kcal/mol) obtained in the kinetic studies. It is also worth noting that the denitrogenation step (the second C−N bond cleavage) is much easier than the ring-opening step (C1−N cleavage). After the first C−N bond cleavage, the following steps are almost barrierless. These results are in sharp contrast with those observed in the activation of pyridine by a titanium alkylidyne complex, where the energy barrier for ring opening (13.0 kcal/mol) was significantly lower than that for denitrogenation promoted by the addition of Me$_3$SiCl (27 kcal/mol)[22].

**Protonolysis studies.** When **3** was treated with hydrochloric acid (HCl) at room temperature, the linear C$_5$ hydrocarbon products were released with the formation of $NH_4Cl$ and (C$_5$Me$_4$SiMe$_3$) TiCl$_3$ (Fig. 5). Gas chromatography–mass spectrometry (GC/MS) and NMR analyses of the volatile products revealed that 1-pentene was formed as a major product (48%) together with 2-pentene (14%) as a minor product. Similarly, the reaction of **5** with HCl yielded 3-methyl-1-pentene (31%), 3-methyl-2-pentene (16%), and some unidentified C$_6$ hydrocarbons accompanied by the formation of $NH_4Cl$ and (C$_5$Me$_4$SiMe$_3$)TiCl$_3$.

In striking contrast, when **3** was hydrolyzed with $H_2O$, the cyclic C$_5$ hydrocarbon product cyclopentadiene was formed exclusively together with [(C$_5$Me$_4$SiMe$_3$)Ti]$_3$($\mu_2$-O)$_3$($\mu_2$-OH)$_3$ and $NH_3$ (Fig. 5). The formation of cyclopentadiene was confirmed by transformation to a structurally characterized ferrocene compound (C$_5$H$_5$)$_2$Fe (52%) through a sequential reaction with NaH and FeCl$_2$, in addition to the GC/MS and NMR analyses of the original volatile product (see Supplementary Methods). The formation of $NH_3$ was proved by reaction with HCl to give $NH_4Cl$ followed by $^1$H NMR and phenol–hypochlorite titration analyses. Analogously, the reaction of **5** with $H_2O$ yielded methylcyclopentadienes, which gave (C$_5$H$_4$Me)$_2$Fe in 61% yield upon sequential reactions with NaH and FeCl$_2$.

It is also worth noting that when the solid residue, which mainly contained [(C$_5$Me$_4$SiMe$_3$)Ti]$_3$($\mu_2$-O)$_3$($\mu_2$-OH)$_3$ formed by hydrolysis of **3** (or **5**) with $H_2O$, was treated with HCl in Et$_2$O at room temperature, the titanium trichloride complex (C$_5$Me$_4$SiMe$_3$)TiCl$_3$ was obtained in 93% yield as dark-orange

crystals. As $(C_5Me_4SiMe_3)TiCl_3$ could be easily transformed to $(C_5Me_4SiMe_3)Ti(CH_2SiMe_3)_3$ by reaction with $LiCH_2SiMe_3$ and subsequently to the hydride complex **1** by hydrogenolysis with $H_2$[38], a titanium-mediated cycle for the hydrodenitrogenation of pyridine to give either linear or cyclic $C_5$ hydrocarbon products could be accomplished as illustrated in Fig. 6.

## Discussion

The present hydrodenitrogenation of pyridines and quinolines by the trinuclear titanium hydride cluster **1** stands in sharp contrast with the previously reported reactions of transition metal hydride complexes with N-heterocycle compounds, in which the cleavage of a C–N bond was not observed[24–37]. The combination of some organometallic complexes with strongly reducing agents or bases such as $LiBEt_3H$, Na/Hg, or $KN(SiMe_3)_2$ was previously reported to induce ring opening (cleavage of one C–N bond) of pyridine, but cleavage of the second C–N bond to extrude the nitrogen atom was not observed[13–19]. In the case of a pyridine ring-opening product generated by metathesis with a titanium alkylidyne species, the denitrogenation was achieved by treatment with an external electrophile $Me_3SiCl$[20–22]. Our studies have revealed both experimentally and computationally that the second C–N bond cleavage is even easier than the first C–N bond scission (pyridine ring opening) at the trinuclear titanium hydride framework, due to the unique synergistic effects of the dynamic and redox-active multiple Ti–H sites. Moreover, the facile formation of **3, 5, 7**, and **9** from **2, 4, 6**, and **8** suggests that the denitrogenation of an aromatic N-heterocycle may not require complete hydrogenation of the aromatic ring. This is different from what was observed previously in the case of solid catalysts, where the complete hydrogenation (saturation) of an aromatic N-heterocycle was believed to be a requisite for C–N bond cleavage (see Fig. 1)[1,2].

The formation of the linear nitrogen-free hydrocarbon products by acidolysis of **3** and **5** with HCl is relatively straightforward and not difficult to understand. However, the formation of a cyclic hydrocarbon product such as cyclopentadiene from the reaction of **3** with $H_2O$ is unusual. This transformation might also reflect the unique behavior of the synergistic trinuclear titanium framework, although further studies are obviously required to clarify the mechanistic details. The formation of a benzene ring through the denitrogenation of pyridine by a titanium alkylidyne complex with the assistance of $Me_3SiCl$ was reported previously[21,22].

In summary, by using a trinuclear titanium heptahydride complex **1**, we have achieved the hydrodenitrogenation of pyridines and quinolines under mild conditions. The reaction takes place sequentially through reduction and deprotonation of a [HC=N] moiety with the release of two molecules of $H_2$ followed by cleavage of the two C–N bonds to give a six-membered metallacycle with a $[Ti_3(\mu_3-N)(\mu_2-H)]$ core structure together with the release of another molecule of $H_2$. Both experimental and theoretical studies have suggested that the second C–N bond cleavage is very fast and easier than the first one (ring opening) at the trinuclear titanium hydride framework. Obviously, the unusually high and unique reactivity of the hydride cluster **1** toward aromatic N-heterocycles is due to the synergistic effects of the dynamic and redox-active multiple Ti–H sites. More intriguingly, the hydrolysis of the denitrogenation products (such as **3** and **5**) with $H_2O$ yields the recyclization products cyclopentadienes, in contrast with the acidolysis with HCl which straightforwardly gives the corresponding linear hydrocarbon products. This work has demonstrated that the denitrogenation of an aromatic N-heterocycle can be achieved through an initial reduction of a C=N bond without hydrogenation of the C=C bonds, and in this

transformation, multimetallic hydrides play a critically important role. These results may provide hints for designing new catalysts for the HDN of aromatic N-heterocycles under milder conditions as well as for other useful chemical transformations[45].

## Methods

**General information**. All reactions were performed under an argon atmosphere. Complex **1** was prepared according to literature procedures[38] and stored in the freezer of a glove box.

**Preparation of 2 from reaction of 1 with pyridine**. Pyridine (15 mg, 0.19 mmol) was slowly added to a hexane solution (5.0 mL) of **1** (100 mg, 0.14 mmol). The solution color immediately changed from dark green to dark purple. The mixture was stirred at room temperature for 10 min, and was then evaporated under vacuum to give a dark-purple crystalline product. Recrystallization in hexane at −33 °C afforded **2** as dark-purple crystals (105 mg, 0.13 mmol, 93%) suitable for X-ray diffraction studies.

**Preparation of 3**. A hexane solution (5.0 mL) of **2** (65 mg, 0.081 mmol) in a 20-mL Schlenk tube equipped with a J. Young valve was stirred at 60 °C for 12 h. After removal of the solvent under vacuum, the resulting residue was dissolved in DME (dimethoxyethane), concentrated, and crystallized at −33 °C to give **3** as dark-green crystals (50 mg, 0.062 mmol, 77% yield) suitable for X-ray diffraction studies.

**Data availability**. The X-ray crystallographic coordinates have been deposited at the Cambridge Crystallographic Database under accession number 1535989, 1535990, 1535991, 1535992, 1535993, 1535994, 1535995, 1536094, and 1538069, respectively. These data can be obtained free of charge from The Cambridge Crystallographic Data Centre via www.ccdc.cam.ac.uk/data_request/cif. All the other data are available from the authors upon reasonable request.

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

## Acknowledgements

This work was supported by a Grant-in-Aid for Young Scientists (B) (No. 26810041) and a Grant-in-Aid for Scientific Research (S) (No. 26220802) from JSPS and a grant from the National Natural Science Foundation of China (No. 21429201). We gratefully appreciate accesses to RICC (RIKEN Integrated Cluster of Clusters) and the Network and Information Centre of Dalian University of Technology for computational resources. We thank Mrs. Akiko Karube for conducting elemental analyses. Dr. Masayoshi Nishiura is gratefully appreciated for his help in X-ray diffraction analyses.

## Author contributions

S.H., T.S., and Z.H.: Conceived and designed the experiments. S.H.: Conducted the experiments. G.L. and Y.L.: Conceived and designed DFT calculations. G.L.: Performed all DFT calculations. S.H., G.L., and Z.H.: Wrote the manuscript. All authors participated in the data analyses and discussions. Z.H. directed the project.

## Additional information

**Competing interests:** The authors declare no competing financial interests.

