## [Peer Review File · Nature Communications]

Reviewer #1 (Remarks to the Author):

The manuscript reported by Hu and co-workers describes a dinitrogenation reaction using a well-defined titanium hydride cluster. What is notable about this N-removal process is that pyridine and quinoline isomers can be the source of the nitrogen. As far as I know, this process is unique in that the nitrogen of the heterocycle is converted to ammonia in the presence of water or acid. The only other example is by Mindiola and co-workers with nitrogen derived from pyridine and using SiMe_3^+ as the electrophile. No examples of ring-opening of quinoline or dinitrogen therefore have been documented. Hence, I am enthusiastic this work be published in Nat. Commun. but pending some major revisions. The authors ignored some important details regarding their work, which must be addressed below.

1. In the abstract, lines 26-27: how exactly will their example help, is one or two steps found critical or could be improved? After reading the paper, I did not find a comment or tow of how their finding will improve the HDN process.
2. Introduction, line 33: after NO_x emissions add "upon combustion of the fuel"
3. Line 36 in the intro: but coal is on the downfall unless countries are reinstating the use of this fuel. Is there a reference backing this up? I doubt coal usage is on the rise!
4. For Fig. 1 the authors should show binding to metal site as well since mentions as an elementary step. They mention this in the text but then ignore it in the figure.
5. Lines 63-64: The authors emphasize over and over again that their system does not require an external reagent to do the ring-opening and N-removal. I am not sure what the benefit of this is over the Mindiola system that uses ClSiMe_3 ? Their system requires input of H_2 to form the hydride and must use a proton source to lose the NH_3 . Either way, the authors should omit this statement because this is not the notable aspect of the paper. They are missing the point, which is that their system mimics the active surface of heterogeneous HDN catalysts! Wolczanski and Mindiola's system can ring-opening pyridine (and derivatives) without external reagents as well. The formation of the nitride in the Mindiola system is facilitated by usage of external ClSiMe_3 .
6. Line 69: Again, the authors emphasize the use of extra reagents to effect N-removal. Instead, I insist the authors focus on the important aspect of this work, which is that they can improve on H_2 usage and uptake since aromatic component in quinoline must be first hydrogenated, and H_2 could be saved if hydrogenolysis of N-C bonds could be selective, rather than wasting it on C-C bonds instead. They are missing the point!
7. Lines 73-74: This statement is absolutely incorrect. Wigley and co-workers observed a metal hydride that can undergo migration to the NC carbon of a side-bound pyridine and result in ring-opening of the ring. However, the pyridine is not derived from a free N-heterocycle, but pre-installed from the coupling of two alkynes and a nitrile under reducing conditions. Please correct this.
8. Line 82: what do the authors mean by "easily", room temperature, mild conditions. The term easy is relative and must be based on something.
9. Line 91, please provide a temperature range.
10. Line 111: The authors should list the isotopologue as 2-d4 instead. There is more than one deuterium.
11. Line 120: the authors should reference some of the work by Santamaria and Mena on trinuclear nitrido systems of Ti with Cp^* .
12. Lines 136-137. This comment is misleading. For ΔH to be negative the bold enthalpy of the bonds being formed must be greater than those being broken! The authors mention what is broken, but never focus on what is formed. What drives this reaction is what is formed, which clearly is overall greater than that was broken.
13. Lines 142-143: The English needs work for this sentence. Maybe rephrase to: rephrase to "Monitoring the conversion of 2 to 3 via ^1H NMR spectroscopy..."
14. Line 144: their process differs significantly from the mechanistic work reported by Mindiola and co-workers, where denitrogenation is harder than ring-opening! No comment is made about this then or later, which seems puzzling since this is something that could help improve the HDN process. Saving H_2 usage could also be another important aspect but the authors do not mention any of this.

15. Lines 145-147: The authors should also list a ΔG to compare since this reaction is relatively slow. Do the authors see any dependence on the rate when the reaction is conducted under an atmosphere of H_2 . Certain studies (in heterogeneous systems) have shown that there is a significance dependence on the rate when higher pressures of H_2 are used. In this work the presence of H_2 might actually inhibit the binding of pyridine, which could be consistent with prior hetero systems. This control reaction should be included.

16. Figure 2: the authors are encouraged to use a different color highlight for the H, because it looks similar to the N.

17. Line 159: this result implies that binding and C-N activation of the substrate might be the RDS. Have the authors considered doing a KIE experiment with d_5 -pyridine to see if the rate is significantly slowed down? I was surprised to see that no kinetic study using d_5 -pyridine was done. It might be possible the authors observe some KIE based on the C-H activation step. In that same token, the authors failed to include a reference by Mindiola and co-workers detailing all the mechanistic studies for dinitrogenation of pyridine with Ti. I am surprised the authors neglected this work since this is the only other example! The authors should read and learn some of these mechanistic details regarding this N-removal reaction. *Organometallics* 2010, 29, 5409-5422.

18. Line 162-163: Generally, quinolines are not more stable but harder to activate given the kinetic hindrance at the quinoline N. They bind poorly to the metal site at the more hindered nitrogen.

19. Line 164: again, provide specifics about what "rapidly means" it is all relative.

20. Figure 3: I am surprised the authors don't even mention that activation of the isoquinoline is regioselective and to the least hindered site! There is an electronic effect regarding this.

21. Line 192: Again, no need to state this non-sense. Instead, this work represents the first example of a well-define metal-hydride species that can denitrogenate N-heterocycles and thus mimic more accurately the active site in the HDN reaction. HDN uses an external substrate so how would this be relevant then? The authors also need to use H_2 to form the hydride precursor, so what?

22. Line 231: again, this contrast the only other example for denitrogenation of pyridine. Please refer to full paper in *Organometallics* by Mindiola and co-workers. There is also a paper in *ICA* by Wolczanski on pyridine ring-opening the authors failed to cite.

23. Line 278: titanium alkylidyne not alkylidene.

24. Lines 291-293: the authors should compare and contrast to the other known homogeneous cycle for pyridine denitrogenation.

25. Line 296: this is redundant to include external reagents, please remove. The system is novel as it stands since it invokes a well-defined metal hydride.

26. Line 202: in the other example known (*Organometallics*, Mindiola), cyclization also occurs to form benzene rings. Please compare and site. If aromaticity is not a driving force, what is the driving force for the N removal, the free energy of NH_3 ? Please note, that NH_3 could in principle be formed in the Mindiola system if such systems were hydrolyzed.

27. Lines 308-311: the authors should emphasize that pre-formation of a hydride, might in principle help in the C-N bond cleavage process without need to hydrogenate C-C bonds and waste H_2 .

In all, this paper is a solid contribution. It is well-written for the most part but some additional experiments are needed as well as correction/omission of some redundant statements. In sum, I think this is solid work that eventually should be published in *Nat. Chem.*

Reviewer #2 (Remarks to the Author):

I have been asked to comment specifically on the crystallographic work in this paper. Overall, I agree that the X-ray data support the interpretations made by the authors. There are a few technical points that should be addressed, which will improve the refinements. Since the authors use the SHELX programs to refine their structures, some of the terminology below refers specifically to SHELX instructions.

(1) In most of the structures (except for 3), the authors should replace "AFIX 33" with "AFIX 137". This allows the H atoms of the methyl groups to rotate around the C—CH₃ bond to fit maxima in the electron density. There are numerous methyl groups in these structures, and the improvement to each refinement is quite significant: wR₂ is reduced by up to 5% in some cases, which leads to quite a significant improvement in the precision of bond distances, etc.

(2) I suggest that the H atoms on the pyridines/quinolones are placed in geometrical positions and refined as riding. I appreciate that the authors do not just assume these positions, and therefore chose to refine them. However, the result in every case is essentially indistinguishable from the standard geometrical positions (I have tried these refinements). So the best approach here is to note that the H atoms appear clearly in the difference Fourier map (nothing is being assumed) and that the positions correspond closely to idealised positions. Using the idealised positions with a riding model for the final refinement minimises the number of parameters and leads to a further improvement in the apparent precision.

(3) The only H atoms being refined should then be the hydrides. These are all very clear in the difference map, and their positions are supported by the fact that they can be refined freely.

(4) The SQUEEZE procedure has been applied to the structure of 6. This is fine, but it should be mentioned in the description in the supporting information. Presumably the solvent is hexane?

Reviewer #3 (Remarks to the Author):

This paper reports the homogeneous HDN of the nitrogen compounds, mechanistic and computational aspects. This paper is rejected for publication.

- 1) There is no need to publish as a communication.
- 2) It is nothing new about the homogeneous HDN of pyridine and quinoline, mechanistic aspect and calculation. The authors should submit other journals (JACS, JOC and JPC).
- 3) Compound: Carbazole should be selected instead of pyridine which was readily decomposed, if the authors study the industrial petroleum refining.
- 4) Catalyst: Is titanium polyhydride complex useful for fundamental importance? Why do they not use Co, Mo, Ni (W) complexes?
- 5) The HDN reaction is initiated with the reaction of the adsorbed molecules and metals on the surface. The authors should study the heterogeneous reaction and calculation instead of one complex and one molecule in a homogeneous system.
- 6) Hydrogen is abundant in the hexane solvent, but the hydrogen supply is important in a practical process. A reviewer thinks that this paper is an old-fashioned study of the mechanistic HDN and active sites of the HDN and HDS catalysts using metal complexes.
- 7) How the addition of HCl or H₂O is meaningful for the HDN reaction?

Reviewer #4 (Remarks to the Author):

The submitted manuscript reports the hydrodenitrogenation of pyridines and quinolines by a trinuclear titanium framework. The authors presented a detailed computational study on the mechanism involved on the reduction of HC=N and cleavage of the two C-N bonds at a trinuclear titanium polyhydride complex. The calculation methodology is adequate, using a functional that included the dispersion effect and basis set big enough to properly describe the reaction mechanism. The article is concise, objective and clear. However, after analysis of the manuscript, I

remains with a couple of questions:

- I'm not convinced that the use of C₅Me₅SiH₃ instead of C₅Me₅SiMe₃ as ligand do not affect significantly the proposed mechanism. The presence of the group SiMe₃ can make more difficult the coordination of the pyridines and quinolies to the titanion molecules and could increase the energy barrier for some transition states. I suggest to the authors to examine the influence of the larger ligand on some representative energetic barriers.

- Is the effect of the solvent important for the energy profile of the mechanism? The table S11 presents the results in presence (and absence) of solvent, but no comments has been done. Please, discuss the effect of the solvent to the overall reaction mechanism.

In conclusion, I recommend the acceptance of the manuscript after minor revision.

Response to Reviewers' Comments

The reviewers' comments are shown in *italic* characters, and our replies are in regular characters.

Reviewer #1

The manuscript reported by Hu and co-workers describes a dinitrogenation reaction using a well-define titanium hydride cluster. What is notable about this N-removal process is that pyridine and quinoline isomers can be the source of the nitrogen. As far as I know, this process is unique in that the nitrogen of the heterocycle is converted to ammonia in the presence of water or acid. The only other example is by Mindiola and co-workers with nitrogen derived from pyridine and using SiMe_3^+ as the electrophile. No examples of ring-opening of quinoline or dinitrogen therefore have been documented. Hence, I am enthusiastic this work be published in Nat. Commun. but pending some major revisions. The authors ignored some important details regarding their work, which must be addressed below.

1. In the abstract, lines 26-27: how exactly will their example help, is one or two steps found critical or could be improved? After reading the paper, I did not find a comment or two of how their finding will improve the HDN process.

Response:

Along with the reduction of the length of the Abstract to less than 150 words according to the format requirements, the last sentence is removed from the Abstract. The comments on how our findings could improve the HDN process are given in the summary section of the main text.

2. Introduction, line 33: after NO_x emissions add “upon combustion of the fuel”

Response:

The revision is made as suggested.

3. Line 36 in the intro: but coal is on the downfall unless countries are reinstating the use of this fuel. Is there a reference backing this up? I doubt coal usage is on the rise!

Response:

The word “coal” is removed to avoid confusing.

4. For Fig. 1 the authors should show binding to metal site as well since mentions as an elementary step. They mention this in the text but then ignore it in the figure.

Response:

It is quite difficult to draw an accurate interaction between the organic molecules and the catalyst surface. To avoid possible confusion, the “catalyst surface” image is now omitted in Fig. 1 (For a similar presentation, see also refs. 1 and 2).

5. Lines 63-64: The authors emphasize over and over again that their system does not require an external reagent to do the ring-opening and N-removal. I am not sure what the benefit of this is over the Mindiola system that uses ClSiMe₃? Their system requires input of H₂ to form the hydride and must use a proton source to lose the NH₃. Either way, the authors should omit this statement because this is not the notable aspect of the paper. They are missing the point, which is that their system mimics the active surface of heterogeneous HDN catalysts! Wolczanski and Mindiola’s system can ring-opening pyridine (and derivatives) without external reagents as well. The formation of the nitride in the Mindiola system is facilitated by usage of external ClSiMe₃.

Response:

The descriptions on the use of an external reagent are appropriately revised as suggested.

6. Line 69: Again, the authors emphasize the use of extra reagents to effect N-removal. Instead, I insist the authors focus on the important aspect of this work, which is that they can improve on H₂ usage and uptake since aromatic component in quinoline must be first hydrogenated, and H₂ could be saved if hydrogenolysis of N-C bonds could be selective, rather than wasting it on C-C bonds instead. They are missing the point!

Response:

As mentioned above, the related descriptions are appropriately revised as suggested.

7. Lines 73-74: This statement is absolutely incorrect. Wigley and co-workers observed a metal hydride that can undergo migration to the NC carbon of a side-bound pyridine and result in ring-opening of the ring. However, the pyridine is not derived from a free N-heterocycle, but pre-installed from the coupling of two alkynes and a nitrile under reducing conditions. Please correct this.

Response:

The description “C-N bond cleavage” is changed to “denitrogenation”.

8. Line 82: what do the authors mean by "easily", room temperature, mild conditions. The term easy is relative and must be based on something.

Response:

To be more specific, “under mild conditions” is added.

9. Line 91, please provide a temperature range.

Response:

The temperature range “(23-25 °C)” is added as suggested.

10. Line 111: The authors should list the isotopologue as 2-d4 instead. There is more than one deuterium.

Response:

The isotopologue “2-d” is changed to “2-d₄” as suggested.

11. Line 120: the authors should reference some of the work by Santamaria and Mena on trinuclear nitrido systems of Ti with Cp*.

Response:

The work by Santamaria and Mena are referenced (refs. 43 and 44) as suggested.

12. Lines 136-137. This comment is misleading. For ΔH to be negative the bond enthalpy of the bonds being formed must be greater than those being broken! The authors mention what is broken, but never focus on what is formed. What drives this reaction is what is formed, which clearly is overall greater than that was broken.

Response:

Here we wanted to focus on the changes of the pyridine skeleton. To avoid misleading, the description is slightly modified. A comprehensive analysis of the whole molecule transformation is given in the computational studies.

13. Lines 142-143: The English needs work for this sentence. Maybe rephrase to: rephrase to "Monitoring the conversion of 2 to 3 via ¹H NMR spectroscopy..."

Response:

The English expression is changed as suggested.

14. Line 144: their process differs significantly from the mechanistic work reported by Mindiola and co-workers, where denitrogenation is harder than ring-opening! No comment is made about

this then or later, which seems puzzling since this is something that could help improve the HDN process. Saving H₂ usage could also be another important aspect but the authors do not mention any of this.

Response:

A comparison with Mindiola's work is given as suggested.

15. Lines 145-147: The authors should also list a deltaG to compare since this reaction is relatively slow. Do the authors see any dependence on the rate when the reaction is conducted under an atmosphere of H₂. Certain studies (in heterogeneous systems) have shown that there is a significance dependence on the rate when higher pressures of H₂ are used. In this work the presence of H₂ might actually inhibit the binding of pyridine, which could be consistent with prior hetero systems. This control reaction should be included.

Response:

“ $\Delta G^\ddagger_{(298.15\text{ K})} = 25.5\text{ kcal/mol}$ ” is provided, as suggested.

The reaction rate of the conversion **2** to **3** under a H₂ atmosphere was not significantly influenced. The related results are mentioned in the text.

16. Figure 2: the authors are encouraged to use a different color highlight for the H, because it looks similar to the N.

Response:

The color of the H atoms in Figs. 2-6 is changed to black to distinguish it from that of the N atoms, as suggested.

17. Line 159: this result implies that binding and C-N activation of the substrate might be the RDS. Have the authors considered doing a KIE experiment with d₅-pyridine to see if the rate is significantly slowed down? I was surprised to see that no kinetic study using d₅-pyridine was done. It might be possible the authors observe some KIE based on the C-H activation step. In that same token, the authors failed to include a reference by Mindiola and co-workers detailing all the mechanistic studies for dinitrogenation of pyridine with Ti. I am surprised the authors neglected this work since this is the only other example! The authors should read and learn some of these mechanistic details regarding this N-removal reaction. Organometallics 2010, 29, 5409-5422.

Response:

The conversion of **1** to **2** was completed almost instantly (within a few minutes) even at -30 °C, and no intermediate was observed. Therefore, it was very difficult to do kinetic studies of this transformation. Instead, we measured the KIE of the conversion of **2** to **3** by using **2** versus **2-d4**. The results are described in the text. Mindiola's work (ref. 22) is also appropriately referenced.

18. Line 162-163: Generally, quinolines are not more stable but harder to activate given the kinetic hindrance at the quinoline N. They bind poorly to the metal site at the more hindered nitrogen.

Response:

As to comparison of quinoline with pyridine, the description "more stable" is changed to "usually more difficult to activate". In this regard, Mindiola's recent work on quinoline ring-opening is cited as ref. 23.

19. Line 164: again, provide specifics about what "rapidly means" it is all relative.

Response:

The reaction time (10 min) was given in Fig. 3. To be more specific, "within 10 min" is added to the text.

20. Figure 3: I am surprised the authors don't even mention that activation of the isoquinoline is regioselective and to the least hindered site! There is an electronic effect regarding this.

Response:

A brief comment on the regioselectivity is added to the text as suggested.

21. Line 192: Again, no need to state this non-sense. Instead, this work represents the first example of a well-define metal-hydride species that can denitrogenate N-heterocycles and thus mimic more accurately the active site in the HDN reaction. HDN uses an external substrate so how would this be relevant then? The authors also need to use H2 to form the hydride precursor, so what?

Response:

As replied above, the related descriptions are appropriately revised as suggested.

22. Line 231: again, this contrast the only other example for denitrogenation of pyridine. Please refer to full paper in Organometallics by Mindiola and co-workers. There is also a paper in ICA by Wolczanski on pyridine ring-opening the authors failed to cite.

Response:

A comparison with Mindiola's work is added as suggested. Wolczanski's paper was cited as ref. 14.

23. *Line 278: titanium alkylidyne not alkylidene.*

Response:

The correction is made as suggested.

24. *Lines 291-293: the authors should compare and contrast to the other known homogeneous cycle for pyridine denitrogenation.*

Response:

Mindiola's work on benzene ring formation is compared as suggested.

25. *Line 296: this is redundant to include external reagents, please remove. The system is novel as it stands since it invokes a well-defined metal hydride.*

Response:

The description "without any extra reducing agents or additives" is removed as suggested.

26. *Line 202: in the other example known (Organometallics, Mindiola), cyclization also occurs to form benzene rings. Please compare and site. If aromaticity is not a driving force, what is the driving force for the N removal, the free energy of NH₃? Please note, that NH₃ could in principle be formed in the Mindiola system if such systems were hydrolyzed.*

Response:

A comparison with Mindiola's work on benzene ring formation is given, as mentioned above.

27. *Lines 308-311: the authors should emphasize that pre-formation of a hydride, might in principle help in the C-N bond cleavage process without need to hydrogenate C-C bonds and waste H₂.*

Response:

The importance of metal hydrides for denitrogenation is briefly described at the last part of the summary.

In all, this paper is a solid contribution. It is well-written for the most part but some additional experiments are needed as well as correction/omission of some redundant statements. In sum, I think this is solid work that eventually should be published in Nat. Chem.

Response:

We gratefully appreciate the reviewer's helpful suggestions and constructive comments on this work.

Reviewer #2

I have been asked to comment specifically on the crystallographic work in this paper. Overall, I agree that the X-ray data support the interpretations made by the authors. There are a few technical points that should be addressed, which will improve the refinements. Since the authors use the SHELX programs to refine their structures, some of the terminology below refers specifically to SHELX instructions.

(1) In most of the structures (except for 3), the authors should replace “AFIX 33” with “AFIX 137”. This allows the H atoms of the methyl groups to rotate around the C—CH₃ bond to fit maxima in the electron density. There are numerous methyl groups in these structures, and the improvement to each refinement is quite significant: wR₂ is reduced by up to 5% in some cases, which leads to quite a significant improvement in the precision of bond distances, etc.

Response:

“AFIX 33” is replaced with “AFIX 137”, as suggested. The changes are reflected in the main text and the new cif files.

(2) I suggest that the H atoms on the pyridines/quinolones are placed in geometrical positions and refined as riding. I appreciate that the authors do not just assume these positions, and therefore chose to refine them. However, the result in every case is essentially indistinguishable from the standard geometrical positions (I have tried these refinements). So the best approach here is to note that the H atoms appear clearly in the difference Fourier map (nothing is being assumed) and that the positions correspond closely to idealised positions. Using the idealised positions with a riding model for the final refinement minimises the number of parameters and leads to a further improvement in the apparent precision.

Response:

The idealized positions with a riding model are used for the refinement of the pyridine/quinoline complexes (2, 4, 6, 7, 8, 9), as suggested. The changes are reflected in the new cif files.

(3) The only H atoms being refined should then be the hydrides. These are all very clear in the difference map, and their positions are supported by the fact that they can be refined freely.

Response:

Yes.

(4) The SQUEEZE procedure has been applied to the structure of 6. This is fine, but it should be mentioned in the description in the supporting information. Presumably the solvent is hexane?

Response:

The use of PLATON SQUEEZE is mentioned in the Supplementary Information (Page 40, line 18), as suggested. The solvent is hexane.

Reviewer #3

This paper reports the homogeneous HDN of the nitrogen compounds, mechanistic and computational aspects. This paper is rejected for publication.

1) There is no need to publish as a communication.

Response:

This is an *Article* submitted to *Nature Communications*.

2) It is nothing new about the homogeneous HDN of pyridine and quinoline, mechanistic aspect and calculation. The authors should submit other journals (JACS, JOC and JPC).

Response:

As described in the manuscript and commented by Reviewer 1, this work represents the first example of HDN of an aromatic N-heterocycle by a well-defined organometallic complex, which could well mimic the active sites (M-H) on the solid catalyst surface at the molecular level. In view of the fundamental interest and importance of carbon–nitrogen bond cleavage in chemical synthesis, petroleum industry, and biological systems, we believe that this work should interest a wide range of audience in the scientific community.

3) Compound: Carbazole should be selected instead of pyridine which was readily decomposed, if the authors study the industrial petroleum refining.

Response:

Pyridines are the most abundant nitrogen-containing component in petroleum, and therefore, have been often used for HDN studies together with quinolines (See refs. 1-4). The content of carbazole in petroleum is much smaller (ref. 1).

4) Catalyst: Is titanium polyhydride complex useful for fundamental importance? Why do they not use Co, Mo, Ni (W) complexes?

Response:

The titanium polyhydride complex reported in this paper is the only molecular metal complex known to date for the HDN of an aromatic N-heterocycle. None of Co, Mo, Ni or W complexes were reported to show activity for the HDN (or cleavage of even one C-N bond) of an aromatic N-heterocycle.

5) *The HDN reaction is initiated with the reaction of the adsorbed molecules and metals on the surface. The authors should study the heterogeneous reaction and calculation instead of one complex and one molecule in a homogeneous system.*

Response:

It is difficult to gain molecular level insights in heterogeneous systems because of the complexity of the active sites on solid catalyst surface. As demonstrated in this work, a well-defined homogeneous organometallic system can provide unprecedented mechanistic details at the molecular level.

6) *Hydrogen is abundant in the hexane solvent, but the hydrogen supply is important in a practical process. A reviewer thinks that this paper is an old-fashioned study of the mechanistic HDN and active sites of the HDN and HDS catalysts using metal complexes.*

Response:

As mentioned above, this work represents the first example of HDN of an aromatic N-heterocycle under mild conditions by a well-defined molecular metal hydride complex, which offers unprecedented insights into the mechanistic details of the activation and denitrogenation of an aromatic N-heterocycle at the molecular level.

7) *How the addition of HCl or H₂O is meaningful for the HDN reaction?*

Response:

Release of the nitrogen-free hydrocarbons and ammonia by hydrogenation (or protonation) is the final process of the industrial HDN reaction. The reaction with HCl or H₂O reported in this paper could be viewed as a mimic of this step. Moreover, the formation of a cyclopentadiene compound has not been observed previously in the conventional HDN reaction. This reaction is unprecedented and unique to the present multimetallic titanium hydride complex.

Reviewer #4

The submitted manuscript reports the hydrodenitrogenation of pyridines and quinolies by a trinuclear titanium framework. The authors presented a detailed computational study on the mechanism involved on the reduction of HC=N and cleavage of the two C-N bonds at a trinuclear titanium polyhydride complex. The calculation methodology is adequate, using a functional that included the dispersion effect and basis set big enough to properly describe the reaction mechanism. The article is concise, objective and clear. However, after analysis of the manuscript, I remains with a couple of questions:

1. I'm not convinced that the use of C₅Me₅SiH₃ instead of C₅Me₅SiMe₃ as ligand do not affect significantly the proposed mechanism. The presence of the group SiMe₃ can make more difficult the coordination of the pyridines and quinolies to the titanion molecules and could increase the energy barrier for some transition states. I suggest to the authors to examine the influence of the larger ligand on some representative energetic barriers.

Response:

The energy profile for the process of **2m**→**C**→**TS_{CD}**→**D** (involving the rate-determining step) with the C₅Me₄SiMe₃ ligand has been calculated to evaluate the substituent influence (Supplementary Figure 60 in SI). The energy difference was found to be less than 2.1 kcal/mol, suggesting that the use of C₅H₄SiH₃ as a model of C₅Me₄SiMe₃ is appropriate in this study.

2. Is the effect of the solvent important for the energy profile of the mechanism? The table S11 presents the results in presence (and absence) of solvent, but no comments has been done. Please, discuss the effect of the solvent to the overall reaction mechanism.

Response:

The relative Gibbs free energies in gas-phase are added in Supplementary Table 11. The difference between ΔG_{gas} and ΔG_{sol} is generally less than 2 kcal/mol, suggesting that the solvation effect has little influence on the overall reaction. Comments are added to the Computational Methods part in SI.

In conclusion, I recommend the acceptance of the manuscript after minor revision.

Response:

We gratefully appreciate the reviewer's comments and recommendation.

Reviewer #1 (Remarks to the Author):

I have read the paper and the comments by the authors (to all reviewers) and was please with their responses. The paper has improved drastically and should now be a great contribution to Nat. Commun. I have no further comments.

Reviewer #2 (Remarks to the Author):

The authors have resolved most of the points that were raised in relation to the crystal structure refinements, which has improved them. They have not in fact changed the descriptions of all non-hydrate H atoms to riding:

Cmpd 3 = 133868_1_data_set_2538769_nvlqm3
Atoms H2, H3, H4, H5 and H6 are still refined freely

Cmpd 5 = 133868_1_data_set_2538772_rvlqm3
Atoms H37, H38, H40 and H41 are still refined freely

Cmpd 7 = 133868_1_data_set_2538765_svlqm3
Atoms H2, H3 and H4 are still refined freely

Cmpd 9 = 133868_1_data_set_2538770_hvlqm3
Atoms H37, H40 and H41 are still refined freely

HOWEVER: there is no reason to insist that they update these files again. The resulting refined structures are just fine, and I consider that all crystallographic work is now acceptable.

Reviewer #4 (Remarks to the Author):

The authors have answered my earlier questions and suggestions, on the computational chemistry part, in a satisfactory way, as a result I now recommend this manuscript to be published in Nature Communication.

Response to Reviewers' Comments

The reviewers' comments are shown in *italic* characters, and our replies are in regular characters.

Reviewer #1

I have read the paper and the comments by the authors (to all reviewers) and was please with their responses. The paper has improved drastically and should now be a great contribution to Nat. Commun. I have no further comments.

Response:

We gratefully appreciate the reviewer's comments, and are pleased that the reviewer is satisfied with our response.

Reviewer #2

The authors have resolved most of the points that were raised in relation to the crystal structure refinements, which has improved them. They have not in fact changed the descriptions of all non-hydride H atoms to riding:

Cmpd 3 = 133868_1_data_set_2538769_nvlqm3

Atoms H2, H3, H4, H5 and H6 are still refined freely

Cmpd 5 = 133868_1_data_set_2538772_rvlqm3

Atoms H37, H38, H40 and H41 are still refined freely

Cpmd 7 = 133868_1_data_set_2538765_svlqm3

Atoms H2, H3 and H4 are still refined freely

Cmpd 9 = 133868_1_data_set_2538770_hvlqm3

Atoms H37, H40 and H41 are still refined freely

HOWEVER: there is no reason to insist that they update these files again. The resulting refined structures are just fine, and I consider that all crystallographic work is now acceptable.

Response:

We thank the reviewer for carefully checking the crystal data. As commented by this reviewer, no change is necessary.

Reviewer #4

The authors have answered my earlier questions and suggestions, on the computational chemistry part, in a satisfactory way, as a result I now recommend this manuscript to be published in Nature Communication.

Response:

We gratefully appreciate the reviewer's comments and recommendation.